# Non-Farm Employment, Farmland Renting and Farming Ability: Evidence from China

**DOI:** 10.3390/ijerph19095476

**Published:** 2022-04-30

**Authors:** Jinning Li, Shunfeng Song, Guanglin Sun

**Affiliations:** 1Xingzhi College, Zhejiang Normal University, Jinhua 321004, China; 2Department of Economics, University of Nevada, Reno, NV 89557, USA; 3School of Finance, Nanjing University of Finance and Economics, Nanjing 210023, China; sunguanglin008@126.com

**Keywords:** non-farm employment, farmland renting-out, farmland renting-in, farming ability

## Abstract

In the process of China’s urbanization, non-farm employment and farmland rental activity are closely correlated. Using data from a survey on rural households in three Chinese provinces, this article examines the relationship between farmland renting activity and non-farm employment with simultaneous equations that consider the farming ability of farmers. Our results are fourfold. First, farmland renting-out promotes non-farm employment, while farmland renting-in reduces non-farm employment. Second, non-farm employment encourages farmland renting-out and decreases farmland renting-in. Third, farming ability increases farmland renting-in but decreases non-farm employment. Fourth, non-farm employment decreases the farming ability of farmers. Based on our empirical findings, we would suggest that the Chinese government further reforms its land system in rural areas, which could better facilitate land-use-right transfer and promote farmland rental market.

## 1. Introduction

As China has been experiencing rapid urbanization over the past decades, urbanization rates increased from 25.84% in 1990, to 36.22% in 2000, 49.95% in 2010, and 63.89% in 2020 (China National Bureau of Statistics). The urbanization process involves both people and land transfers from the rural sector to the urban sector. The former is evident by the mass peasant migration, such as a rural-migrant population of about 290.77 million in 2019. The latter includes transferring land ownership from collectives to the state and reclassifying suburban rural areas into urban areas. Many studies have investigated the push and pull factors of rural-urban migration [1,2,3], the contributions to urbanization made by migrant workers [2], challenges faced by migrant workers [3], and land issues related to compensation on land ownership transfers and settlements paid to the farmers who lost their land due to urbanization [3,4]. Additionally, some researchers paid attention to the pressure on farmland loss, inefficient land-use patterns, food security, food sovereignty and de-agrarianization caused by urban expansion [5,6,7,8], and readily support the preservation of farmland in urbanization [9,10].

Land rental activities in rural areas are an important issue that is understudied. When farmers move to cities, they leave their land behind in their villages. If a migrant worker’s family lacks sufficient labor to farm its own land, it either leaves the land vacant or rents it out to other farmers. Rural households that have surplus labor or want to enjoy scale effects rent more land for larger-scale of production. A farmland rental market emerges in rural areas, and this is the research focus of our paper.

Several interesting questions arise with the land rental market in rural areas. What affects a rural family’s decision to rent out its land? Would the rental activities further encourage farmers to migrate to urban areas and seek non-farm jobs? Recent research shows farming ability has an important influence on farmland rental activity and non-farm employment [11,12,13,14,15]. Do simultaneous relationships exist among non-farm employment, farmland renting, and farming ability? Do the market behaviors differ between the renting-out and the renting-in groups? 

This paper attempts to answer these questions and aims to enrich the literature on rural migration and land market. There are great differences in farmland transfer mechanisms between rural-to-urban and rural-to-rural. The scope of this research is limited to rural-to-rural land transfers. Since the farmland is owned by the village in China, in this paper, farmland transfers refer to the right-to-use only, with the land ownership remaining to the village. 

This article uses survey data collected from rural areas in Zhejiang, Henan, and Shaanxi provinces in 2017. It employs a simultaneous equation system to examine the determinants of non-farm employment, farmland renting, and farming ability. Our empirical results show that: (1) farmland renting-out promotes non-farm employment while farmland renting-in reduces non-farm employment; (2) non-farm employment encourages farmers to rent out farmland and decreases the chances for farmers to rent in farmland; (3) farming ability decreases non-farm employment, while increases farmland renting-in; (4) non-farm employment decreases farming ability.

The rest of this paper is organized as follows. In Section 2, we discuss the research background, conduct a literature review, and propose hypotheses. Section 3 develops the simultaneous equations, describes the data, and defines the variables. Section 4 presents the empirical results, performs robustness checks, and discusses identification issues. Section 5 provides conclusions and policy implications.

## 2. Background, Literature Review, and Hypotheses

### 2.1. Background

Over the past decades, a mass of farmers has moved to urban areas. On the one hand, China’s rapid urbanization and industrialization call for rural-to-urban migration of labor [16]. Pulling factors in urban areas (such as higher wages, better education, and more opportunities) also attract rural laborers to flow into non-farm sectors. On the other hand, pushing factors in rural areas (such as lack of farmland, low agricultural productivity, high tax burden before 2004, and poor quality of life) encourage farmers to move into cities [2]. Both the pulling and pushing factors have caused millions of migrants to move or drift into non-farm sectors, often leaving farmland behind in their villages.

However, farmers are mandated to work on their farmland allocated by their villages. Otherwise, according to the Land Management Law of China, village committees could retrieve farmland from rural households if the land stays vacant over two consecutive years [17]. Therefore, farmers face choices in their decision-making. They could seek non-farm jobs in urban areas, continue to work on their farmland, rent out or rent in farmland, or let their farmland vacant and face the risk of being retrieved.

Decisions on these choices are not independent, due to the fact that seeking jobs in urban areas often implies that they may not have sufficient labor to work on farmland and thus have to either rent out farmland or let the farmland vacant. Likewise, staying in rural areas may suggest that farmers have surplus labor; it would be more productive if they could rent in some farmland to expand the scale of production. Also, in farmers’ decision-making, farming ability could be a factor, which itself could diminish with time that farmers have moved to urban areas and over generations. The above discussions raise the following questions: Will non-farm employment necessarily lead to farmland rent-out or rent-in? Does farmland rent-out or rent-in have an impact on non-farm employment? What are the relationships among non-farm employment, farmland renting, and farming ability? Answering these questions will shed important insights on better understanding the rural farmland rental market, migration behavior, farmers’ non-farm employment, poverty reduction in rural areas, and farmland production efficiency [18,19]. 

### 2.2. Literature Review

Our literature review is two-fold. One is about the relationship between non-farm employment and farmland renting activities. Some researchers suggested that non-farm employment shapes farmland renting activities [20,21]. From the perspective of non-farm employment’s stability, Du and Li [22] showed that a permanent non-farm employment of a farmer has a positive impact on farmland transfer, while a temporary non-farm employment of a farmer is not conducive to farmland transfer. From the perspective of household resource allocation, Salvioni [23] showed that the opportunities of non- farm job may affect the decisions of peasant households in household resource allocation. In other words, in the lack of non- farm labor chances, pluriactivity is not available to solve the problem of low farming incomes, when farmers are more dependent on income from farming. This will make farmers more inclined to expand by renting in land from others. The conclusion was supported by Vranken [24] and Akter [25], who suggested farmland is usually rented out to farmers not involving in non-farm jobs, and the lack of non-farm labor opportunities encourages a farming household to rent in more land or rent out less land, so opportunities of non-farm employment are identified as critical factors affecting land rental activities of rural households especially in transition countries. As for the measurement of non-farm employment, we notice that previous studies measured non-farm employment differently. For example, Kung [20] and De Janvry [26] used the share of non-farm income, Che [27] used the number of non-farm workers in a household, and Willmore et al. [28] used the share of a household’s non-farm labor. Other studies suggested farmland renting activities affect non-farm employment. The exiting literature shows renting out farmland promotes non-farm employment of farmers [29]. It is especially true in government-oriented transfers of farmland [30]. Government-oriented transfers of farmland often happen in China, when farmers losing farmland have to seek for non-farm jobs to raise their family income, but not necessarily in other countries, so a large amount of relevant literature is concentrated in China.

The other fold is about the role played by farming ability in farmland rental activity and non-farm employment. In terms of the relationship between farming ability and farmland renting activity, a number of previous studies concluded that farmers with greater farming ability are more likely to rent in farmland than those with poor farming ability [11,12,13,14,15]. For instance, Akter [25] suggested that farmland is transferred to those with relatively smaller holding, but greater ability to make productive use of land. This conclusion suggests that farmland rental markets facilitate agricultural production efficiency, as more able farmers rent in farmland from those with less ability. As far as the relationship between farming ability and non-farm employment is concerned, farming households usually seek to combine non-farm and on-farm generating activities in order to optimize their total income and establish a suitable balance between the amount of labor needed for farming activities and the labor required for non-farming activities [31]. In this way, farmers with great farming abilities can be allocated for farming jobs in peasant households instead of non-farm jobs. Besides that, Mishra and Goodwin [32] suggested that more farming experience corresponds to less non-farm employment, which likely reflects the fact that farming experience builds farming-specific human capital. As a result, farming ability and farming income are raised, which makes them less likely to be employed non-farm. 

The main findings from previous studies are (a) non-farm employment encourages farmland rent-out; (b) farmland rent-out promotes farmers’ non-farm employment; (c) farming ability plays an important role in non-farm employment and farmland rental activity. However, previous studies only focused on one-way causal relationships, although some studies tried to use instrumental variables to correct the biased estimates caused by endogeneity [20,33]. Particularly, farming ability of farmers was ignored as an endogenous variable, which renders inconsistent and biased results. 

Therefore, this paper makes two contributions to the existing literature. One is to consider farming ability as an endogenous variable. Another is the use of a simultaneous equation system in examining the mutual relationships among non-farm employment, farmland renting activities, and farming ability. In other words, we take the decisions of non-farm employment, farmland rental activity, and farming ability as a whole to estimate the mutual effects, rather than use separate equations to examine one-way causal relationships. Doing so, we can obtain results that are more robust, reliable, and consistent. 

### 2.3. Hypotheses Development

Historically speaking, China’s household contract responsibility system in rural areas has played a huge role in economic development. It was the first step of China’s economic reforms, which not only allowed farmers to decide what to grow and thus improved production efficiency, but also freed up rural labors which can better meet the increasing demand for labor in China’s industrialization and urbanization. However, such system created fragmented farmland for farmers and made rural households responsible for their land allocated by villages. On the one hand, the small-scale agricultural cultivation can create difficulties for both individual farms and farming communities [34], for example, it impedes the introduction of new technologies, farming machines, and novel production models, limiting the labor and machine efficiency [35,36,37,38,39]. On the other hand, China’s farmers have a deep local attachment to their farmland, which makes them want to stay home, which could cause an income loss and a labor shortage in urban sectors.

A possible solution is to develop a farmland rental market in rural areas. In the market, some rural households can rent-out their farmland, so they can free up more time and engage in part- or full-time non-farm employment to increase income. Other farmers can rent-in farmland to increase their production scale, especially for farmers with greater farming ability [11,12,13,14,15]. A number of previous studies have proved that farmland transfer is an important way to improve farmland production efficiency [10,15,19,39]. With the development of farmland rental market, rural resources could be better used, since renting out farmland frees up some farmers for non-farm employment while renting in farmland allows other farmers to increase production scale, especially those who have greater farming ability. Therefore, this study proposes the following hypothesis:

**Hypothesis** **1** **(H1).**
*Renting out farmland promotes the non-farm employment of farmers while renting in farmland decreases the non-farm employment of farmers.*


China’s industrialization and urbanization in the past decades provided a vast number of opportunities for farmers to engage in non-farm employment. The higher wages in urban areas also attracted many farmers to migrate into cities [2,3]. Among the migrant workers, most are young and middle-aged farmers who are the main labor force for farming activities [40]. One consequence is that some rural households became labor shortage. For these households, as an “economic man”, renting out part or all of their farmland could be a one-stone-two-birds choice under the current system and policy. In addition, the past commentators have indicated that taking up off-farm work would result in a decline in the technical efficiency of farming production [31], which is enable farmers involving in non-farm jobs to rent out their farmland for rent income. On the one hand, renting out farmland would allow farmers to collect some symbolic rent and receive government agricultural subsidies. On the other hand, renting out farmland helps farmers prevent their farmland from being confiscated by their villages. As we discussed earlier, under the current Chinese law, villages have the right to reclaim farmland if it has been vacant for more than two years. Based on these two reasons, farmers who engaged in non-farm employment would be more likely to rent their farmland out. For the same reasons, the lack of farming labor caused by non-farm employment limits the room on renting in farmland. Accordingly, this study proposes the following hypothesis:

**Hypothesis** **2** **(H2).**
*The non-farm employment of farmers promotes the rent-out of farmland but reduces the rent-in of farmland.*


The existing literature shows that rural households with greater farming ability are more likely to rent in farmland [11,12,14,15,41]. A greater farming ability suggests a higher marginal product of labor. It has two implications for farmer’s decision-making. One is a direct impact. A higher marginal product suggests a bigger return from farming on land, causing a stronger demand for farmland. This, in turns, decreases farmland rent-out but increases farmland rent-in. The other is an indirect impact. A higher marginal agricultural product suggests a higher opportunity cost of taking non-farm jobs, giving farmers less incentive to engage in non-farm employment. Therefore, this study proposes the following two hypotheses:

**Hypothesis** **3** **(H3).**
*A household’s farming ability decreases farmland rent-out but increases farmland rent-in.*


**Hypothesis** **4** **(H4).**
*A household’s farming ability decreases non-farm employment.*


Off-farm employment may accelerate exits from farming [42]. Thus, farmers’ farming ability declines with the length of their non-farm employment. Physically, farm and non-farm jobs could demand quite differently. After a long time of non-farm employment, a farmer could become incompetent doing farming work any longer. Technologically, farming changes over time, including what machines to run, what fertilizers and pesticides to use, and even what and when to produce. If farmers have worked in urban sectors for a long time, they could become unfamiliar with the current cultivating skills and their farming ability declines. Based on the above analysis, the following hypothesis is proposed:

**Hypothesis** **5** **(H5).**
*Non-*
*farm employment decreases a household’s farming ability.*


Figure 1 depicts the possible causal relationships between non-farm employment, farmland renting, and farming ability. We aim to provide statistical evidence on them in our analysis below.

## 3. Simultaneous Equation Model, Data, and Variables

### 3.1. Simultaneous Equation Model

Given the above discussions on non-farm employment, farmland renting, and farming ability, simultaneous equation models are suitable for our empirical analysis. We specify the following simultaneous equations.
*NONFARM* = a_10_ + a_11_*RENT* + a_12_*ABILITY* + a_13_X_1_ + ε_1_(1)
*RENT* = a_20_ + a_21_*NONFARM* + a_22_*ABILITY* + a_23_X_2_ + ε_2_(2)
*ABILITY* = a_30_ + a_31_*NONFARM*+a_32_X_3_ + ε_3_(3)
where *NONFARM* denotes non-farm employment; *RENT* denotes farmland renting activities, that refers to either rent-out or rent-in depending on sample groups. Farmland renting activities is measured by the ratio of rent-out or rent-in land amount to the household’s initial farmland; *ABILITY* represents farming ability; vectors X_j_ (j = 1, 2, 3) include all household-level characteristics, village-level characteristics, and policy-level characteristics; a_ij_ (i = 1, 2, 3 and j = 0, 1, 2, 3) are parameters to be estimated; and ε_j_ (j = 1, 2, 3) are error terms. 

### 3.2. Data

Data used in this paper come from the survey on rural households in Zhejiang, Henan, and Shaanxi provinces. The survey was conducted and administrated by the Rural Research Center at Zhejiang Normal University during the 2017 Spring Festival holiday. Since migrant farmers largely return to their hometowns and stay at home during this major Chinese holiday, the survey was able to obtain comprehensive and accurate information on rural households, with or without migrant farmers. As the largest developing country in the world, the levels of economic development and urbanization are quite uneven among different regions in China. We expect that non-farm employment, farmland rental activities, and farmland resources per capita also vary across regions. The survey selected Zhejiang Province in the eastern region, Henan Province in the central region, and Shaanxi Province in the western region. We understand that they may not be able to fully represent China’s three main regions. But given the resource constraint, the survey was unable to include more provinces.

To better ensure the representativeness of the sample, in each sampled province, the survey used a stratified random sampling procedure. Specifically, the survey randomly selected six counties from each province, five townships from each county, two villages from each township, and 4–7 households from each village depending on the village size. When sampling, the survey tried to keep a balance between villages that are suburbs of cities and villages that are far from cities. The respondents of the survey include three groups: households who were renting in farmland from other rural families, households who were renting out their farmland, and households who did neither. Questions asked in the survey were about non-farm employment, farmland rental activities, farming ability, household situation, household head situation, village situation, and policy on farmland. Most information was collected at the household level; some was collected at the village level, such as the village location and the average income of households in the village. After dropping the observations with missing values on the key variables or obvious errors, we have 1001 observations from 180 different villages. The data on per capita GDP at prefecture-level city are derived from the China Statistical Yearbook.

Figure 2 shows that the sample is quite evenly distributed across the three provinces, with 36.06%, 29.67%, and 34.27%, respectively, for Zhejiang, Shaanxi, and Henan. Figure 3 shows that about half of the households (47.85%) in the survey did not engage in farmland renting activities, one third of rural households (33.57%) rented out their farmland, and less than 20% of rural households (18.58%) rented in farmland from other farmers. Our data suggest that farmland renting activities are not uncommon in rural areas, although China’s farmland rental market has a big room for future development [43]. Figure 4 shows the sample shares of households based on the number of migrant workers for non-farm jobs. We observed that about two third of households have migrant workers who engage in non-farm employment, indicating that non-farm jobs are common for rural households. We also observed that more households have 2 or more migrant workers than those with single migrant worker, suggesting that rural households often migrant jointly.

### 3.3. Variables

In this article, the core variables include non-farm employment, farmland renting, and farming ability, which are all endogenous. Non-farm employment is measured by the share of non-farm income as used in the literature [20,44]. For robustness tests, the study also uses the number of non-farm workers in a household [27,45] to replace the share of non-farm income. Farmland renting is measured by the farmland renting ratio, which is the proportion of farmland that is rented relative to the total amount of farmland initially endowed to the household by the village [20]. Also, we replace it with farmland renting incidence late for robustness tests. The farming ability of the household is measured with the Likert scale by farmer’s self-evaluation according to their farming experiences.

Exogenous variables are those about household-level characteristics, village-level characteristics, and policy-level characteristics. Household-level characteristics include household head’s situation (physical condition, education level, cognitive level of farmland tenure, age, spouse, the training of farming skills), per capita farmland area, agricultural fixed assets, average farming years of household members, and dependency ratio. We consider agricultural fixed assets since the residual value of the used specialized agricultural fixed assets is low [46], which will increase the opportunity cost of giving up farming and engaging in non-farm employment. To measure the dependency ratio, we divided the number of non-working age members by the number of working age members in a household. Following the international practice and considering that most farmers still work on farms until 64, we define working age as the ages of 15-64. To measure dependency ratio, we divided the number of non-working age members by the number of working age members in a household. Village-level characteristics include the average income of households in the village, the location of the village, and the stability of farmland tenure, which is measured by the frequency of farmland adjustment by its village. Policy-level characteristics include farmland subsidies, new farmer insurance, and farmland title. It is worth mentioning that some variables are ordinal variables such as head physical condition, education level, cognitive level of farmland tenure, and the stability of farmland tenure; some are nominal (dummy) variables, such as spouse, farmland title, farmland subsidy, and new farmer insurance. In the article, the rent of farmland is not included as a variable since rent is not a critical factor in China’s farmland rental market. According to the Land Management Law of China, village committees could retrieve farmland from rural households if the land stays vacant over two consecutive years [17]. Therefore, rural households often rent out their farmland to their neighbors, relatives, or friends at a very low price or even for free to avoid from confiscating their farmland by their villages or disqualifying them for a farmland subsidy from the government. 

Table 1 presents descriptive statistics about the variables used in the study for both rent-out and rent-in groups. *RENT* denotes farmland renting ratio, *NONFARM* represents non-farm income share, and *ABILITY* means farming ability. We observe some significant differences in *NONFARM*, *RENT*, and *ABILITY* between the both groups. First, the mean non-farm income share is much higher for the renting-out group (81.98%) than that for the renting-in group (55.55%). This correlation between non-farm income shares and farmland rental activities is not a surprise. The mean of farmland renting ratio is also higher for the rent-in group (2.43) than that for the rent-out group (0.76), suggesting that farmers want to enjoy scale effect in their production when they rent in farmland. As for farming ability (*ABILITY*), the mean value is greater for the rent-in group than that for the rent-out group. It shows that farmers in the rent-in group are of better farming ability than those in the rent-out group. The mean value of *SKILL* is higher for the rent-in group (3.67) than that for the rent-out group (3.01) since the farmers with farming training have greater farming ability, thus are more likely to rent in farmland. Among the explanatory and control variables, most are similar between the rent-out and rent-in groups. As expected, households in the rent-in group have more agricultural fixed assets and per capita farmland area.

## 4. Empirical Analyses

In this section, we perform empirical analyses on factors that affect non-farm income share, farmland renting ratio, and farming ability. In our analyses, we separate the rent-out group from the rent-in group.

We estimated Equations (1)–(3) individually and simultaneously. The estimated coefficients from OLS are generally smaller than those from multi-equation system since OLS ignores the relationships among the equations. For the multi-equation system, if any endogenous independent variables are included in the equation system, the two-stage least squares’ (2SLS) estimation results of each equation are consistent. However, they are not the most effective since 2SLS ignores the possible correlation among the disturbances of equations. Both OLS and 2SLS belong to the single equation estimation method, which separately estimates each equation in the simultaneous equations. Our main estimation method is three-stage least squares (3SLS), which treats all of the equations as a whole. In the first stage, the reduced form of simultaneous equation system is estimated. Then, the 2SLS estimates of all equations in the simultaneous equation system are obtained by the fitting of all endogenous variables. Once the 2SLS parameters are calculated, the residuals of each equation can be used to estimate the variances and covariances between equations. In the third stage, the parameter estimators of the generalized least square method are obtained, so the most effective estimation results could be obtained. Therefore, the 3SLS can better address the problem of endogeneity and correlations among error terms in estimating.

In the following discussions, this paper focuses on the estimation results from 3SLS, although it reports the estimation results from OLS and 2SLS for references. To save space, we only present and discuss the results on *NONFARM*, *RENT*, and *ABILITY*. The results on other explanatory and control variables are available upon request.

### 4.1. Regression Results and Discussion

#### 4.1.1. Determinants of Non-Farm Income Share

Panel A in Table 2 presents the regression results on the share of non-farm income (*NONFARM*) based on Equation (1), for the rent-out and rent-in groups, respectively. For the rent-out group, the 3SLS coefficient of RENT is statistically positive at the 1% significance level. This indicates that farmland rent-out ratio increases the non-farm income share. Since renting out farmland frees up time for non-farm employment and thus promotes non-farm employment. In some circumstances, to some extent farmland is compulsively rented out in China, especially in the government-oriented transfer model, when farmers have to seek non-farm jobs to increase their family income. Therefore, renting farmland out is regarded as an important aspect affecting non-farm employment. For the rent-in group, the 3SLS coefficient of RENT is statistically negative at the 5% significance level. This result shows farmland rent-in ratio lifts the share of non-farm employment. We interpret it as evidence that renting in farmland indeed increases the scale of agricultural production, which, in turn, increases the demand for farming labor. It further leads to the lack of migrant workers available for non-farm employment in households and decreases the share of non-farm income. In addition, farmers renting in farmland may obtain scale returns, which lifts the opportunity costs of non-farm employment. Therefore, they may give up some chances of non-farm employment, then both the probability and share of non-farm employment may fall. These results support our hypothesis *H1**: Renting out farmland promotes the non-farm employment of farmers while renting in farmland decreases the non-farm employment of farmers.*

Panel A in Table 2 shows that the coefficients of *ABILITY* are significantly negative for both rent-out and rent-in groups, suggesting that farming ability decreases the share of non-farm income. In detail, farmers with great ability can obtain a more marginal return from farming than others, thus compared to others, those farmers are more likely to engage in farming jobs and less likely to involve in non-farm jobs. This finding verifies our hypothesis *H4: A household’s farming ability decreases non-farm employment*.

#### 4.1.2. Determinants of Farmland Renting Ratio

Panel B in Table 2 presents the regression results on farmland renting ratio based on Equation (2). For the rent-out group, we observed a positive and significant 3SLS coefficient of *NONFARM*, showing that the share of non-farm income promotes the rent-out farmland activities. A higher share of non-farm income often suggests a higher non-farm employment share for a rural family. In turn, a higher non-farm employment share could suggest a lack of farming labor, causing the household to rent out more farmland. Moreover, vacant farmland for over two consecutive years will be retrieved by the village in China, so farmers without enough farming labor have to rent out all or part of their farmland. So non-farm employment promotes renting out farmland, which is consistent with previous literature [47]. For the rent-in group, in contrast, we observed a statistically negative 3SLS coefficient of NONFARM, showing that non-farm employment (represented here by the share of non-farm income) decreases the rent-in ratio of farmland, probably since non-farm employment reduces farmers’ capacity to rent in farmland for a larger scale of agricultural production. The higher the share of non-farm income, the more time or labor the households spend on non-farm jobs. Correspondingly, less time will be spent on farming activities, so they are more incentive to rent out their farmland and less incentive to rent in farmland from others. These results support our hypothesis H2: The non-farm employment of farmers promotes the rent-out of farmland and reduces the rent-in of farmland.

Panel B in Table 2 shows a negative but insignificant result of *ABILITY* on *RENT* for the rent-out group, suggesting that farming ability has little impact on farmers’ rent-out activities. This weak relationship between farming ability and rent-out farmland activity could suggest that rural households rent out farmland, not due to poor farming ability but due to other reasons, e.g., high risks and low income from farming, especially lacking the access to non-farm employment (hypothesis 2). The conclusion is supported by much existing literature [22,47]. In short, some farmers are reluctant to cultivate their farmland for various reasons but are under pressure of being retrieved farmland by villages, so they select to rent out their farmland. if the land stays vacant over two consecutive years. For the rent-in group, however, the coefficient of *ABILITY* is significantly positive, which shows that farming ability promotes the rent-in ratio of farmland. This finding is not surprising, since higher farming ability suggests better labor productivity, which increases the marginal product of farmland, calling for more land input in production. These results partly support our hypothesis *H3: A household’s farming ability decreases the rent-out of farmland but promotes the rent-in of farmland.*

#### 4.1.3. Determinants of Farming Ability

Panel C in Table 2 shows the regression results on farming ability based on Equation (3). For both rent-out and rent-in groups, we found that the coefficients of *NONFARM* are statistically negative at the 1% significance level, indicating that the share of non-farm income decreases farming ability. In reality, farmers with non-farm experience tend to select renting out farmland and non-farm employment in future [32,48], which pushes them further away from farming activities. This makes them worse at farming. This empirical finding confirms our hypothesis *H5: Non-farm employment decreases a household’s farming ability*. Panel A, B, and C in Table 2 have the same sample size.

### 4.2. Robustness Checks

The statistical results for the relationships among non-farm employment, farmland renting, and farming ability are intuitively plausible in light of the literature on agricultural economics. But how robust are they across alternative specifications of the variables? To the best of our knowledge, non-farm employment and farmland renting could be measured in different ways as mentioned earlier, and we choose the following two measurements for our robustness tests.

#### 4.2.1. The Number of Migrant Workers

In addition to the share of non-farm income, non-farm employment could be measured by the number of migrant workers in a household [27,45]. We regard the number of migrant workers as the measurement of non-farm employment in the robustness test, and the symbol is *MIRATE*, keeping other variables unchanged. The robustness check results are shown in Table 3.

Panel A in Table 3 presents the results on the number of migrant workers in a rural household for both rent-out and rent-in groups. The results confirm the findings we observed in Table 2 Panel A. For the rent-out group, the 3SLS estimated coefficient of *RENT* is significantly positive, indicating that the rent-out ratio of farmland frees up rural labors and thus increases the number of migrant workers. For the rent-in group, the 3SLS estimated coefficient of *RENT* is significantly negative, suggesting that renting in farmland demands more rural labor and thus decreases the number of migrant workers. As for *ABILITY*, the estimated coefficients are significantly negative for both groups, confirming that farming ability decreases the number of migrant workers. These results strongly support the finding that the renting out of farmland promotes non-farm employment, renting-in of farmland decreases non-farm employment, and farming ability decreases non-farm employment.

Panel B in Table 3 presents the results on the farmland renting ratio. For the rent-out group, the results show significant and positive coefficients of *MIRATE*, indicating that the number of migrant workers increases the rent-out ratio of farmland. For the rent-in group, the results show that significant but negative coefficients of *MIRATE*, suggesting that the number of migrant workers reduces the rent-in ratio of farmland. The results on *ABILITY* show that farming ability has little impact on renting-out activities but enables farmers to rent-in more farmland in their agricultural production. These results are consistent with our earlier findings that non-farm employment increases farmland rent-out of but decreases farmland rent-in, and farming ability increases farmland rent-in.

Panel C in Table 3 shows negative but significant results for all models and for both rent-out and rent-in groups. These results further provide evidence that non-farm employment decreases farming ability. Panel A, B, and C in Table 3 have the same sample size.

#### 4.2.2. Farmland Renting Incidence

Following Deaton et al. [49], we measure farmland renting activities by farmland renting incidence (*INCIRENT*), with *INCIRENT* taking a value of 1 if a rural household rents out or rents in farmland, or it takes a value of 0 otherwise. Therefore, in our regression analysis below, sample points which show neither rent-out nor rent-in are double used as the reference group, respectively for the rent-out and rent-in groups.

The results of our robustness checks are presented in Table 4. Panel A shows that *INCIRENT* has a positive estimated coefficient for the rent-out group, while it has a negative estimated coefficient for the rent-in group, both are statistically significant. Therefore, we conclude that farmland renting incidence helps to raise the share of non-farm income, but it makes a rural household become less dependent on non-farm income if it rents in farmland from other villagers. We won’t interpret these results undesirable, since renting out farmland could free up rural labor for more non-farm employment and earn a higher total income, while renting in farmland could expand the scale of agricultural production and raise rural household’s overall income. The coefficients of *ABILITY* are significantly negative for both groups, which shows that farming ability always decreases the share of non-farm income. Panel A confirms that our previous results on the determinants of non-farm income share are robust.

The 3SLS results in Panel B of Table 4 show that a rural household with a higher share of non-farm income becomes more (less) likely to rent out (in) farmland. We consider this finding expected, since a higher share of non-farm income indicates a larger share of non-farmland employment among a family’s labor. In turns, when a rural household lacks of farming labor, due to non-farm employment, a natural choice is to rent out its farmland. Renting in farmland is the other side of the same story. The estimated coefficients of *ABILITY* are significant and positive for the rent-in group but not significant for the rent-out group. Therefore, farming ability only helps farmers when they decide to rent in farmland; it does not matter for farmers who choose to rent out their farmland. Panel B confirms that our previous results on the determinants of farmland renting decision are robust.

Panel C in Table 4 shows the estimation results of farming ability. For both rent-out and rent-in groups, based on the significantly negative estimated coefficients of *NONFARM*, the results indicate that non-farm employment decreases farming ability, confirming our earlier finding. Panel A, B, and C in Table 4 have the same sample size.

Panel C in Table 4 shows the estimation results of farming ability. For both rent-out and rent-in groups, based on the significantly negative estimated coefficients of *NONFARM*, the results indicate that non-farm employment decreases farming ability, confirming our earlier finding. Panel A, B, and C in Table 4 have the same sample size.

## 5. Conclusions

Over the past decades, a mass of farmers has moved to urban areas and become migrant workers in China’s rapid process of urbanization and industrialization. However, to some extent, farmers do not want to part with their farmland allocated by their villages, which also happened in other countries [47]. Otherwise, according to the Land Management Law of China, village committees could retrieve farmland from rural households if the land stays vacant over two consecutive years. Therefore, farmers face choices in their decision-making. They could seek non-farm jobs in urban areas, continue to work on their farmland, rent out or rent in farmland, or let their farmland vacant and face the risk of being retrieved. Decisions on these choices are not independent, since seeking jobs in urban areas often implies that they may not have sufficient labor to work on farmland and thus have to either rent out farmland or let the farmland vacant. Likewise, staying in rural areas may suggest that farmers have surplus labor; it would be more productive if they could rent in some farmland to expand the production scale. Also, in farmers’ decision-making, farming ability could be a factor, which itself could diminish with time that farmers have moved to urban areas and over generations.

Using survey data from Zhejiang, Henan, and Shaanxi provinces in 2017, this paper has investigated the simultaneous relationships among non-farm employment, farmland renting, and farming ability. We obtained three main conclusions. First, farmland rent-out promotes non-farm employment while farmland rent-in decreases non-farm employment. Farming ability decreases non-farm employment. Second, non-farm employment encourages farmland rent-out whereas it decreases farmland rent-in. Farming ability promotes farmland rent-in but it has no effect on farmland rent-out. Third, non-farm employment decreases farming ability.

### Policy Implications

Based on our empirical results, we would propose the following two policy implications. First, the current fragmented land allocation system in rural China results in a small per capita farmland and causes low agricultural production efficiency. An effective farmland rental market could provide a win-win solution for all farmers. On the one hand, renting out farmland would free up more rural laborers so that they can work in the urban sector. On the other hand, renting in farmland would enable farmers to expand their production scale, apply new technology, and improve production efficiency. Therefore, it is important for the Chinese government to promote farmland rental markets in rural areas. Second, under the current system, agricultural land is owned collectively by villages. Farmers have the right to use, but they are not allowed to sell any farmland individually. We would propose that China could treat rural households as “shareholders” of their collectively-owned land and allow them to sell part of their shares at least back to their villages. Such a reform could help migrant workers better settle down in cities, as rural migrants could use the money to buy homes or cover the down payment when they buy houses in urban areas. In reality, many migrant workers have worked in cities for many years, and they will continue to live and work in cities. Deregulating the rural land system and improving the farmland market could help China narrow the rural-urban divide and promote a common prosperity development.

## Figures and Tables

**Figure 1 ijerph-19-05476-f001:**
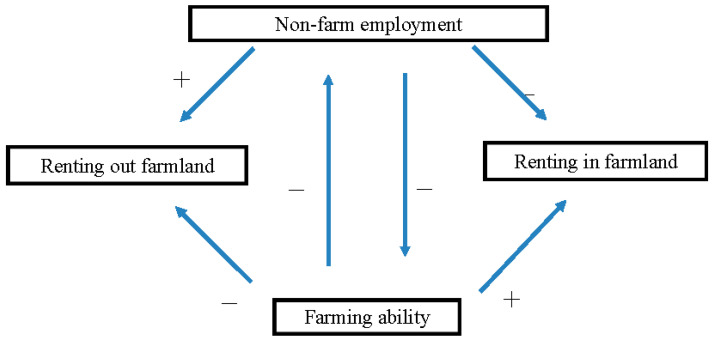
Possible causal relationships among non-farm employment, farmland renting, and farming ability.

**Figure 2 ijerph-19-05476-f002:**
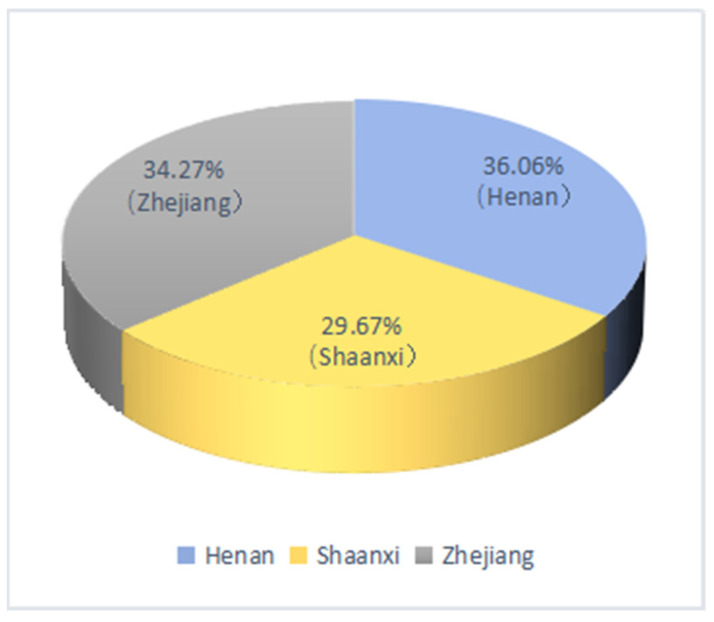
Share of samples, by region.

**Figure 3 ijerph-19-05476-f003:**
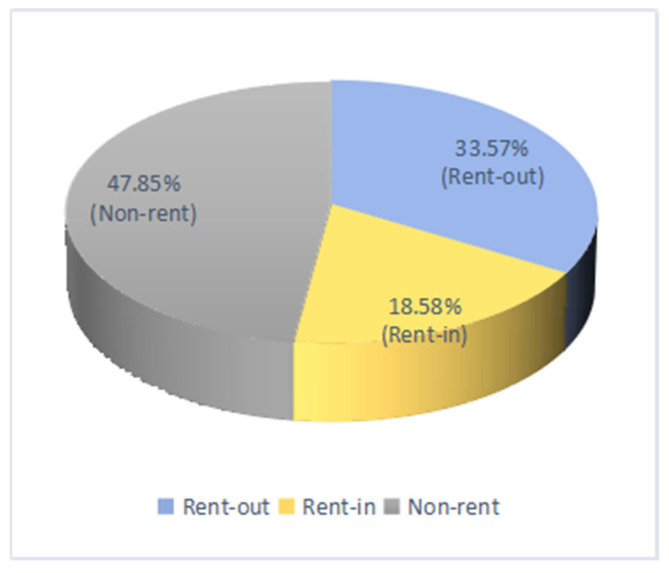
Share of samples, by renting activity.

**Figure 4 ijerph-19-05476-f004:**
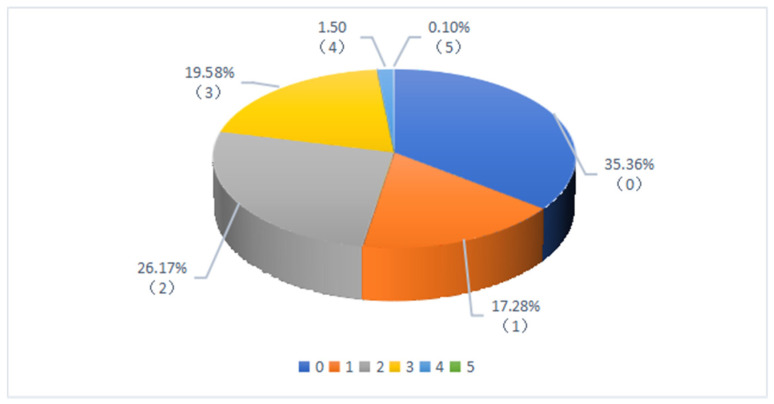
Share of samples, by the number of migrant workers in a household.

**Table 1 ijerph-19-05476-t001:** Descriptive statistics on the variables.

Symbols	Variables	Measurements(Unit)	Rent-Out Group		Rent-In Group	
			Mean	SD	Mean	SD
NONFARM	Non-farm employment	Non-farm income share(non-agricultural income/total household’s income) (%)	81.98	28.51	55.55	33.4
RENT	Farmland renting	Farmland renting ratio(rental farmland /total household’s initial farmland) (%)	0.76	0.28	2.43	5.35
ABILITY	Farming ability	Likert five scale	3.01	1	3.67	0.86
PHY	Head physical condition	Likert five scale	3.74	0.64	3.83	0.96
EDU	Head educational level	Illiterate = 1, elementary = 2 school, middle school = 3, high school = 4, college or above = 5	2.69	0.83	2.75	0.95
FYEAR	Years of farming (year)	Average farming years of household members	15.20	14.40	14.50	11.68
SPOUSE	Spouse	yes = 1, no = 0	0.96	0.21	0.96	0.19
DASSET	Agricultural fixed asset	Agricultural fixed asset value per household (10,000 yuan)	0.92	2.66	4.08	9.72
AGE	Head age		54.47	10.77	52.42	9.98
SKILL	Farm skills training	Length of farming skills training of household head (months)	0.28	2.11	0.70	1.72
DERATE	Dependency ratio	The number of non-working age members/the number of working age members in a household (%)	0.56	0.65	0.51	0.49
ACRPER	Per capital farmland acres	Total farmland acres/the number of family members (mu)	0.86	0.73	2.65	3.93
COGNI	Cognitive level of farmers about farmland tenure	How many rights a farmer think he has to his(her) farmland	1.95	0.95	1.87	0.86
STAB	Stability of farmland tenure	Adjusting frequency of farmland by its village	2.22	0.82	2.13	0.91
VPOSI	Location of the village	The distance of the village from the nearest town (km)	1.57	5.44	1.93	7.54
VINCO	Village income level	Per household income in village (10,000 yuan)	2.74	1.85	3.21	3.17
SUBSIDY	Farmland subsidy policy	yes = 1, no = 0	0.650	0.480	0.750	0.43
INSURE	New farmer insurance policy	yes = 1, no = 0	0.880	0.330	0.700	0.46
TITLE	Farmland title	yes = 1, no = 0	0.710	0.460	0.660	0.48
GDP	Per capita GDP (10,000 yuan)		4.51	0.57	4.97	0.66

**Table 2 ijerph-19-05476-t002:** Determinants of non-farm income share, farmland renting ratio, and farming ability.

Variables	Rent-Out Group			Rent-In Group		
	(1) OLS	(2) 2SLS	(3) 3SLS	(4) OLS	(5) 2SLS	(6) 3SLS
Panel A: Determinants of non-farm income share						
RENT	0.010 ***	0.054 ***	0.404 ***	−0.439 **	−1.718 *	−1.846 **
	(0.002)	(0.017)	(0.014)	(0.177)	(0.954)	(0.782)
ABILITY	−0.036 ***	−0.052	−0.464 ***	−0.539 ***	−0.412 *	−1.114 ***
	(0.064)	(0.128)	(0.111)	(0.192)	(0.238)	(0.192)
control	*yes*	*yes*	*yes*	*yes*	*yes*	*yes*
Panel B: Determinants of farmland renting ratio						
NONFARM	4.224 ***	7.150	9.950 ***	−0.089 ***	−0.142	−0.337 ***
	(1.288)	(7.153)	(3.159)	(0.034)	(0.112)	(0.078)
ABILITY	−4.316 **	−3.280	−0.614	0.012	0.022 *	0.093 **
	(2.068)	(3.249)	(2.229)	(0.060)	(0.075)	(0.065)
control	*yes*	*yes*	*yes*	*yes*	*yes*	*yes*
Panel C: Determinants of farming ability						
NONFARM	−0.154 ***	−0.789 ***	−0.907 ***	−0.097 **	−0.818 ***	−0.816 ***
	(0.042)	(0.171)	(0.148)	(0.043)	(0.164)	(0.156)
control	*yes*	*yes*	*yes*	*yes*	*yes*	*yes*
*N*	336	336	336	186	186	186

*, **, *** denote significance at the 10%, 5% and 1% level respectively.

**Table 3 ijerph-19-05476-t003:** Robustness check results of simultaneous equations: with the number of non-farm workers.

Variables	Rent-Out Group			Rent-In Group		
	(1) OLS	(2) 2SLS	(3) 3SLS	(4) OLS	(5) 2SLS	(6) 3SLS
Panel A: Determinants of the number of non-farm workers						
RENT	0.270 *	0.274 *	0.542 ***	0.012	−0.479 *	−0.729 ***
	(0.139)	(0.143)	(0.069)	(0.069)	(0.287)	(0.143)
ABILITY	−0.003 *	−0.075	−0.104 **	−0.167 *	−0.532 *	−1.315 ***
	(0.067)	(0.169)	(0.157)	(0.102)	(0.427)	(0.279)
control	*yes*	*yes*	*yes*	*yes*	*yes*	*yes*
Panel B: Determinants of farmland renting ratio						
	(1) OLS	(2) 2SLS	(3) 3SLS	(4) OLS	(5) 2SLS	(6) 3SLS
MIRATE	0.031	1.584 **	1.690 ***	0.031	−1.018	−1.309 ***
	(0.022)	(0.623)	(0.469)	(0.085)	(0.787)	(0.329)
ABILITY	−0.042	−0.226	0.111	0.266 **	0.783	1.502 ***
	(0.027)	(0.283)	(0.257)	(0.113)	(0.658)	(0.438)
control	*yes*	*yes*	*yes*	*yes*	*yes*	*yes*
Panel C: Determinants of farming ability						
MIRATE	−0.146 ***	−0.479 ***	−0.683 ***	−0.088	−0.390	−0.614 **
	(0.042)	(0.163)	(0.143)	(0.057)	(0.323)	(0.261)
control	*yes*	*yes*	*yes*	*yes*	*yes*	*yes*
N	336	336	336	186	186	186

*, **, *** denote significance at the 10%, 5% and 1% level respectively.

**Table 4 ijerph-19-05476-t004:** Robustness check results of simultaneous equations (with farmland renting incidence).

	Rent-Out Group			Rent-In Group		
	(1) OLS	(2) 2SLS	(3) 3SLS	(4) OLS	(5) 2SLS	(6) 3SLS
Panel A: Determinants of non-farm income share						
RENTINCI	0.102 **	0.647 **	0.898 ***	−0.386 ***	−2.590 ***	−3.011 ***
	(0.046)	(0.262)	(0.198)	(0.128)	(0.670)	(0.566)
ABILITY	−0.234 ***	−0.806 ***	−0.884 ***	−0.306 ***	−0.976 ***	−1.206 ***
	(0.049)	(0.170)	(0.162)	(0.063)	(0.294)	(0.242)
control	*yes*	*yes*	*yes*	*yes*	*yes*	*yes*
Panel B: Determinants of farmland renting incidence						
NONFARM	0.046 *	0.689	1.002 ***	−0.020 *	−0.121 *	−0.257 ***
	(0.027)	(0.505)	(0.241)	(0.012)	(0.068)	(0.041)
ABILITY	−0.120 ***	0.634	0.890	0.080 ***	0.154	0.309 ***
	(0.037)	(0.443)	(0.252)	(0.019)	(0.112)	(0.091)
control	*yes*	*yes*	*yes*	*yes*	*yes*	*yes*
Panel C: Determinants of farming ability						
NONFARM	−0.080 ***	−0.050	−0.090 **	−0.099 ***	−0.174 **	−0.187 **
	(0.024)	(0.079)	(0.075)	(0.024)	(0.080)	(0.073)
control	*yes*	*yes*	*yes*	*yes*	*yes*	*yes*
N	815	815	815	665	665	665

*, **, *** denote significance at the 10%, 5% and 1% level respectively.

## Data Availability

The study did not report any data.

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
