# Peer review of "Non-Farm Employment, Farmland Renting and Farming Ability: Evidence from China"

_ijerph, 2022, doi:10.3390/ijerph19095476_

Round 1

Reviewer 1 Report

First of all, I would like to highlight the importance of the analysis of the topic addressed and the convenience of the authors making complementary and subsidiary realistic proposals to avoid the shortcomings that other similar processes have had in other territories. In these, urbanization has caused an extreme de-agrarianisation which has put at serious risk the social, economic and territorial organization, as well as food sovereignty and independence. The processes studied here have many parallels with those that occurred in European countries decades ago, although with extraordinary quantitative differences. However, the review of the literature on this subject is excessively brief and many of the main bibliographical references of both European and American authors are missing. The bibliographical references focus too much on studies referring to China and are very limited in terms of the analysis of similar processes in other territories. On the other hand, the methodology used is quite simplistic, although it is well explained.
Consequently, the results contribute very little to the advancement of knowledge on this problem since they are limited to repeating the results of the mathematical formulas but lack qualitative interpretation. The results are quite obvious and, in reality, they are limited to reiterating hypotheses and ideas contrasted long ago in other territories and areas of knowledge.
The graphic information is excessively basic.
In my opinion, the final proposal is highly debatable: “Based on our empirical results, we would propose the following policy recommendations. First, to further facilitate farmers to migrate into cities and to improve agricultural productivity for farmers who stay in rural areas, the Chinese government could relax more regulations on rural land transfers" (rows 477-480).    

Reviewer 2 Report

Abstract

Lines 8 to 10…. “This article focuses on how to interact between farmland renting activity and non-farm employment considering the farming ability of farmers, that exploits simultaneous equations strategy based on a data set”

  • This statement is unclear. Kindly reformulate to enhance clarity

Lines 10 to 14… “Our results are fourfold. First, farmland renting-out promotes non-farm employment, while farmland renting-in reduces non-farm employment. Second, non-farm employment encourages farmland renting-out and decreases farmland renting-in. Third, farming ability increases farmland renting-in but decreases non-farm employment. Fourth, non-farm employment decreases the farming ability of farmers.”

  • What is the policy/economic implications of these identified associations?

1.0 Introduction

Lines 52 and 53…”Non-farm employment encourages farmers to rent out farmland and decreases farmers to rent in farmland”

  • Consider replacing with .. “Non-farm employment encourages farmers to rent out farmland and decreases the chances for farmers to rent in farmland”.

2.0 Background, literature review, and hypotheses

2.1 Background

Lines 69 and 70…. “Both the pulling and pushing factors have caused millions of migrants into non-farm sectors, often leaving farmland behind in their villages”

  • Consider replacing with …. “Both the pulling and pushing factors have caused millions of migrants to move/drift into non-farm sectors, often leaving farmland behind in their villages.

2.2 Hypotheses development

Lines 168 and 169… “H3: A household’s farming ability decreases farmland rent-out but decreases farmland rent-in”

  • This statement is confusing/unclear….kindly check and edit

  1. Simultaneous equation model, data and variables

3.1 Simultaneous equation model

Line 210… “RENT denotes farmland renting;”

  • Renting out or renting in?

3.3 Variables

Lines 270 to 272…. “To measure dependency ratio, we divided the number of non-working age members by the number of working age members in a household.”

  • How do you define ‘non-working age’ and ‘working age’?

Comments on results

Since the section was labelled (with the heading) results, I was hoping to see a separate section on discussion. The section was a mere report of the correlations between the indicators, with nothing being said about the policy and economic implications, and how the findings of this study refute or conform with claims from earlier/previous studies.

Round 2

Reviewer 1 Report

In my opinion, the authors have made great efforts to improve their manuscript according to the indications and suggestions that I have made. Some underlying issues (conceptual and methodological) remain unresolved but are not important enough to prevent the acceptance and publication of this text.